# ~~Convolutions Attention MLPs~~
# Patches Are All You Need? 🤷

**Asher Trockman, J. Zico Kolter**[1]
*Carnegie Mellon University and *[1]*Bosch Center for AI*

**Reviewed on OpenReview:** *https://openreview.net/forum?id=rAnB7JSMXL*

## Abstract

Although convolutional neural networks have been the dominant architecture for computer vision for many years, Vision Transformers (ViTs) have recently shown promise as an alternative. Subsequently, many new models have been proposed which replace the self-attention layer within the ViT architecture with novel operations (such as MLPs), all of which have also been relatively performant. We note that these architectures all share a common component—the patch embedding layer—which enables the use of a simple isotropic template with alternating steps of channel- and spatial-dimension mixing. This raises a question: is the success of ViT-style models due to novel, highly-expressive operations like self-attention, or is it at least in part due to using patches? In this paper, we present some evidence for the latter: specifically, we propose the ConvMixer, an extremely simple and parameter-efficient fully-convolutional model in which we replace the self-attention and MLP layers within the ViT with less-expressive depthwise and pointwise convolutional layers, respectively. Despite its unusual simplicity, ConvMixer outperforms the ViT, MLP-Mixer, and their variants for similar data set sizes and parameter counts, in addition to outperforming classical vision models like ResNet. We argue that this contributes to the evidence that patches are sufficient for designing simple and effective vision models. Our code is available at https://github.com/locuslab/convmixer.

## 1 Introduction

For many years, convolutional neural networks have been the dominant architecture for deep learning systems applied to computer vision tasks. But recently, architectures based upon *Transformer* models, *e.g.*, the so-called Vision Transformer architecture (Dosovitskiy et al., 2020), have demonstrated compelling performance in many of these tasks, often outperforming classical convolutional architectures, especially for large data sets. An understandable assumption, then, is that it is only a matter of time before Transformers become the dominant architecture for vision domains, just as they have for language processing. In order to apply Transformers to images, however, the representation had to be changed: because the computational cost of the self-attention layers used in Transformers would scale quadratically with the number of pixels per image if applied naively at the per-pixel level, the compromise was to first split the image into multiple "patches", linearly embed them, and then apply the transformer directly to this collection of patches.

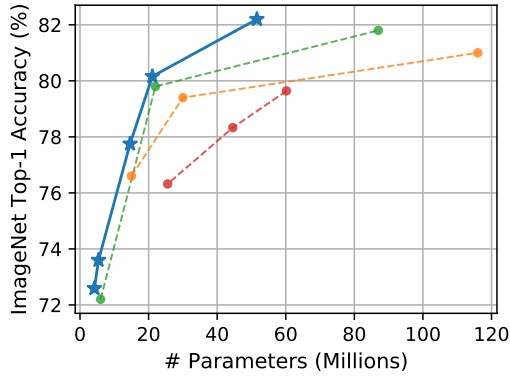

Figure 1: Acc. *vs.* params., trained & tested on ImNet-1k; ResNets newly-trained (same procedure as ConvMixers).

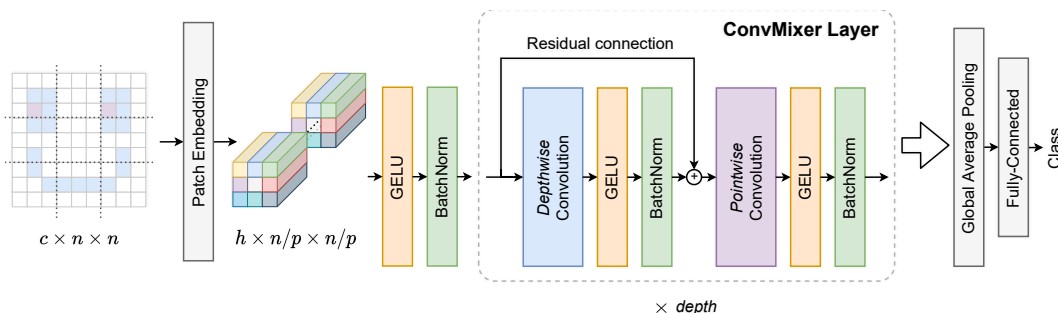

Figure 2: ConvMixer uses "tensor layout" patch embeddings to preserve locality, and then applies $d$ copies of a simple fully-convolutional block consisting of *large-kernel* depthwise convolution followed by pointwise convolution, before finishing with global pooling and a simple linear classifier.

```python
import torch.nn as nn

class Residual(nn.Module):
    def __init__(self, fn):
        super().__init__()
        self.fn = fn

    def forward(self, x):
        return self.fn(x) + x

def ConvMixer(dim, depth, kernel_size=9, patch_size=7, n_classes=1000):
    return nn.Sequential(
        nn.Conv2d(3, dim, kernel_size=patch_size, stride=patch_size),
        nn.GELU(),
        nn.BatchNorm2d(dim),
        *[nn.Sequential(
                Residual(nn.Sequential(
                    nn.Conv2d(dim, dim, kernel_size, groups=dim, padding="same"),
                    nn.GELU(),
                    nn.BatchNorm2d(dim)
                )),
                nn.Conv2d(dim, dim, kernel_size=1),
                nn.GELU(),
                nn.BatchNorm2d(dim)
        ) for i in range(depth)],
        nn.AdaptiveAvgPool2d((1,1)),
        nn.Flatten(),
        nn.Linear(dim, n_classes)
    )
```

Figure 3: Implementation of ConvMixer in PyTorch; see Appendix E for more implementations.

Many subsequent works have modified the architecture of the ViT, replacing self-attention with novel operations and making other small changes, all of which have been relatively performant. These architectures follow a common and very simple "template": they are isotropic, maintaining equal size and resolution throughout the network, and apply alternating steps of spatial and channel mixing. They also all use patch embeddings, which moves all downsampling to the beginning of the network and enables the simple, isotropic mixing design.

In this work, we explore the question of whether, fundamentally, the strong performance of vision transformers may result more from this patch-based representation and its simplifying consequences for architecture design, than from the use of novel and highly-expressive operations such as self-attention and MLPs. We develop a very simple convolutional architecture which we dub the "ConvMixer" due to its similarity to the recently-proposed MLP-Mixer (Tolstikhin et al., 2021). This architecture is similar to the Vision Transformer (and MLP-Mixer) in many respects: it directly operates on patches, it maintains an equal-resolution-and-size representation throughout all layers, it does no downsampling of the representation at successive layers, and it separates "channel-wise mixing" from the "spatial mixing" of information. But unlike the Vision Transformer and MLP-Mixer, our architecture does all these operations via only standard convolutions. As

depthwise and pointwise convolution are less expressive than self-attention and MLPs respectively, we believe this suggests that the patch-based isotropic mixing architecture is a powerful primitive that works well with almost any choice of well-behaved mixing operations.

The chief result we show in this paper is that this ConvMixer architecture, despite its extreme simplicity (it can be implemented in 280 characters of dense PyTorch code; see Appendix E), outperforms both "standard" computer vision models such as ResNets of similar parameter counts *and* some corresponding Vision Transformer and MLP-Mixer variants, even with a slate of additions intended to make those architectures more performant on smaller data sets. Importantly, this is despite the fact that *we did not design our experiments to maximize accuracy nor speed*, in contrast to the models we compared against. Our results suggest that, at least to some extent, the *patch representation itself* may be a critical component to the "superior" performance of newer architectures like Vision Transformers. While these results are naturally just a snapshot, and more experiments are required to exactly disentangle the effect of patch embeddings from other factors, we believe that this provides a strong "convolutional-but-patch-based" baseline to compare against for more advanced architectures in the future.

## 2 A simple model: ConvMixer

Our model, dubbed *ConvMixer*, consists of a patch embedding layer followed by repeated applications of a simple fully-convolutional block. We maintain the spatial structure of the patch embeddings, as illustrated in Fig. 2. Patch embeddings with patch size $p$ and embedding dimension $h$ can be implemented as convolution with $c_{\text{in}}$ input channels, $h$ output channels, kernel size $p$, and stride $p$:

$$z_0 = \text{BN}\left(\sigma\{\text{Conv}_{c_{\text{in}} \rightarrow h}(X, \texttt{stride=}p, \texttt{ksize=}p)\}\right) \tag{1}$$

The ConvMixer block itself consists of depthwise convolution (*i.e.*, grouped convolution with groups equal to the number of channels, $h$) followed by pointwise (*i.e.*, kernel size $1 \times 1$) convolution. As we will explain in Sec. 3, ConvMixers work best with unusually large kernel sizes for the depthwise convolution. Each of the convolutions is followed by an activation and post-activation BatchNorm:

$$z_l' = \text{BN}\left(\sigma\{\text{ConvDepthwise}(z_{l-1})\}\right) + z_{l-1} \tag{2}$$
$$z_{l+1} = \text{BN}\left(\sigma\{\text{ConvPointwise}(z_l')\}\right) \tag{3}$$

After many applications of this block, we perform global pooling to get a feature vector of size $h$, which we pass to a softmax classifier. See Fig. 3 for an implementation of ConvMixer in PyTorch.

**Design parameters.** An instantiation of ConvMixer depends on four parameters: (1) the "width" or hidden dimension $h$ (*i.e.*, the dimension of the patch embeddings), (2) the depth $d$, or the number of repetitions of the ConvMixer layer, (3) the patch size $p$ which controls the internal resolution of the model, and (4) the kernel size $k$ of the depthwise convolutional layer. We name ConvMixers after their hidden dimension and depth, like ConvMixer-$h/d$. We refer to the original input size $n$ divided by the patch size $p$ as the *internal resolution*; note, however, that ConvMixers support variable-sized inputs.

**Motivation.** Our architecture is based on the idea of *mixing*, as in Tolstikhin et al. (2021). In particular, we chose depthwise convolution to mix *spatial locations* and pointwise convolution to mix *channel locations*. A key idea from previous work is that MLPs and self-attention can mix distant spatial locations, *i.e.*, they can have an arbitrarily large receptive field. Consequently, we used convolutions with an unusually large kernel size to mix distant spatial locations.

While self-attention and MLPs are theoretically more flexible, allowing for large receptive fields and content-aware behavior, the inductive bias of convolution is well-suited to vision tasks and leads to high data efficiency. By using such a standard operation, we also get a glimpse into the effect of the patch representation itself in contrast to the conventional pyramid-shaped, progressively-downsampling design of convolutional networks.

Table 1: Models trained and evaluated on $224 \times 224$ ImageNet-1k only. See more in Appendix A.

| | | | | Current "Most Interesting" **ConvMixer** Configurations *vs.* Other Simple Models | | | |
|---|---|---|---|---|---|---|---|
| Network | Patch Size | Kernel Size | # Params ($\times 10^6$) | Throughput (img/sec) | Act. Fn. | # Epochs | ImNet top-1 (%) |
| ConvMixer-1536/20 | 7 | 9 | 51.6 | 134 | G | 150 | 81.37 |
| ConvMixer-768/32 | 7 | 7 | 21.1 | 206 | R | 300 | 80.16 |
| ConvMixer-1536/20 | 14 | 9 | 52.3 | 538 | G | 150 | 78.92 |
| ResNet-152 | – | 3 | 60.2 | 828 | R | 150 | 79.64 |
| DeiT-B | 16 | – | 86 | 792 | G | 300 | 81.8 |
| ResMLP-B24/8 | 8 | – | 129 | 181 | G | 400 | 81.0 |

## 3  Experiments

**Training setup.** We primarily evaluate ConvMixers on ImageNet-1k classification without any pretraining or additional data. We added ConvMixer to the `timm` framework (Wightman, 2019) and trained it with nearly-standard settings for the common training procedure from this library: we used RandAugment (Cubuk et al., 2020), mixup (Zhang et al., 2017), CutMix (Yun et al., 2019), random erasing (Zhong et al., 2020), and gradient norm clipping in addition to default `timm` augmentation. We used the AdamW (Loshchilov & Hutter, 2018) optimizer and a simple triangular learning rate schedule. Due to limited compute, *we did virtually no hyperparameter tuning* on ImageNet and trained for fewer epochs than competitors. Consequently, our models could be over- or under-regularized, and the accuracies we report likely underestimate the capabilities of our model.

**Results.** A ConvMixer-1536/20 with 52M parameters can achieve 81.4% top-1 accuracy on ImageNet, and a ConvMixer-768/32 with 21M parameters 80.2% (see Table 1). Wider ConvMixers seem to converge in fewer epochs, but are more memory- and compute-hungry. They also work best with large kernel sizes: ConvMixer-1536/20 lost $\approx 1\%$ accuracy when reducing the kernel size from $k = 9$ to $k = 3$ (we discuss kernel sizes more in Appendix A, B, & C). ConvMixers with smaller patches are substantially better in our experiments, similarly to Sandler et al. (2019); we believe larger patches require deeper ConvMixers. With everything held equal except increasing the patch size from 7 to 14, ConvMixer-1536/20 achieves 78.9% top-1 accuracy but is around $4\times$ faster. We trained one model with ReLU to demonstrate that GELU (Hendrycks & Gimpel, 2016), which is popular in recent isotropic models, isn't necessary.

**Comparisons.** Our model and ImageNet1k-only training setup closely resemble that of recent patch-based models like DeiT (Touvron et al., 2020). Due to ConvMixer's simplicity, we focus on comparing to only the most basic isotropic patch-based architectures adapted to the ImageNet-1k setting, namely DeiT and ResMLP. Attempting a fair comparison with a standard baseline, we trained ResNets using exactly the same parameters as ConvMixers; while this choice of parameters is suboptimal (Wightman et al., 2021), it is likely also suboptimal for ConvMixers, since we did *no hyperparameter tuning* aside from our recent adoption of hyperparameters from Wightman et al. (2021) for some models (presented separately in Appendix A). Looking at Table 1 and Fig. 1, ConvMixers achieve competitive accuracies for a given parameter budget: ConvMixer-1536/20 outperforms both ResNet-152, ResMLP-B24, and DeiT-B despite having substantially fewer parameters. ConvMixer-768/32 uses just a third of the parameters of ResNet-152, but is similarly accurate. Note that unlike ConvMixer, the DeiT and ResMLP results involved hyperparameter tuning, and when substantial resources are dedicated to tuning ResNets, including training for twice as many epochs, they only outperform an equivalently-sized ConvMixer by $\approx 0.2\%$ (Wightman et al., 2021). However, ConvMixers are substantially slower at inference than the competitors, likely due to their smaller patch size; hyperparameter tuning and optimizations could narrow this gap. For more discussion and comparisons, see Table 2 and Appendix A.

**Hyperparameters.** For almost all experiments presented in the main text, we used only one set of "common sense" hyperparameters for the regularization methods. Recently, we adapted hyperparameters from the A1 procedure in Wightman et al. (2021), published after our work, which were better than

our initial guess, *e.g.*, giving +0.8% for ConvMixer-1536/20, or 82.2% top-1 accuracy (see Appendix A). However, we note that such optimal ResNet hyperparameters are likely not optimal for ConvMixers.

**Additional experiments.** We present additional ImageNet experiments in Appendix B; notably, we provide more evidence for the advantage of large-kernel convolutions. We also performed smaller-scale experiments on CIFAR-10, where ConvMixers achieve over 96% accuracy with as few as 0.7M parameters, demonstrating the data efficiency of the convolutional inductive bias. Details of these experiments are presented in Appendix C.

## 4 Related work

**Isotropic architectures.** Vision transformers have inspired a new paradigm of "isotropic" architectures, *i.e.*, those with equal size and shape throughout the network, which use patch embeddings for the first layer. These models look similar to repeated transformer-encoder blocks (Vaswani et al., 2017) with different operations replacing the self-attention and MLP operations. For example, MLP-Mixer (Tolstikhin et al., 2021) replaces them both with MLPs applied across different dimensions (*i.e.*, spatial and channel location mixing); ResMLP (Touvron et al., 2021a) is a data-efficient variation on this theme. CycleMLP (Chen et al., 2021), gMLP (Liu et al., 2021a), and vision permutator (Hou et al., 2021), replace one or both blocks with various novel operations. These are all quite performant, which is typically attributed to the novel choice of operations. In contrast, Melas-Kyriazi (2021) proposed an MLP-based isotropic vision model, and also hypothesized patch embeddings could be behind its performance. ResMLP tried replacing its linear interaction layer with (small-kernel) convolution and achieved good performance, but kept its MLP-based cross-channel layer and did not explore convolutions further. As our investigation of ConvMixers suggests, these works may conflate the effect of the new operations (like self-attention and MLPs) with the effect of the use of patch embeddings and the resulting isotropic architecture.

After our investigation, Liu et al. (2022) proposed an architecture similar to ConvMixer, the isotropic ConvNeXt. Similarly to our work, they provide evidence that the success of ViTs comes from design choices other than the use of self-attention, such as patches; however, ConvMixer goes a step further and eliminates even the MLPs, which suggests that neither of the original ViT operations are crucial to the success of the more general architecture design. Further, Yu et al. (2022) replaced self-attention with a simple pooling operation and demonstrated its effectiveness; they also argued this supports the effectiveness of the ViT template. In contrast, our work suggests the template is even more general, not even requiring MLPs.

A study predating vision transformers investigates isotropic (or "isometric") MobileNets (Sandler et al., 2019), and even implements patch embeddings under another name. Their architecture simply repeats an isotropic MobileNetv3 block. They identify a tradeoff between patch size and accuracy that matches our experience, and train similarly performant models (see Appendix A, Table 2). However, their block is substantially more complex than ours; simplicity and motivation sets our work apart.

**Patches aren't all you need?** Several papers have increased vision transformer performance by replacing standard patch embeddings with a different stem: Xiao et al. (2021) and Yuan et al. (2021a) use a standard convolutional stem, while Yuan et al. (2021b) repeatedly combines nearby patch embeddings. However, this conflates the effect of using patch embeddings with the effect of adding convolution or similar inductive biases *e.g.*, locality. We attempt to focus on the use of patches.

**CNNs meet ViTs.** Many efforts have been made to incorporate features of convolutional networks into vision transformers and vice versa. Self-attention can emulate convolution (Cordonnier et al., 2019) and can be initialized or regularized to be like it (d'Ascoli et al., 2021); other works simply add convolution operations to transformers (Dai et al., 2021; Guo et al., 2021), or include downsampling to be more like traditional pyramid-shaped convolutional networks (Wang et al., 2021). Conversely, self-attention or attention-like operations can supplement or replace convolution in ResNet-style models (Bello et al., 2019; Ramachandran et al., 2019; Bello, 2021). While all of these attempts have been successful in one way or another, they are orthogonal to this work, which aims to emphasize the effect of the architecture common to most ViTs by showcasing it with a less-expressive operation.

## 5 Conclusion

We presented ConvMixers, an extremely simple class of models that independently mixes the spatial and channel locations of patch embeddings using only standard convolutions. We also highlighted that using large kernel sizes, inspired by the large receptive fields of ViTs and MLP-Mixers, provides a substantial performance boost. While neither our model nor our experiments were designed to maximize accuracy or speed, *i.e.*, we did not search for good hyperparameters, ConvMixers outperform the Vision Transformer and MLP-Mixer, and are competitive with ResNets, DeiTs, and ResMLPs.

We provided evidence that the increasingly common "isotropic" architecture with a simple patch embedding stem is itself a powerful template for deep learning. Patch embeddings allow all the downsampling to happen at once, immediately decreasing the internal resolution and thus increasing the effective receptive field size, making it easier to mix distant spatial information. Our title, while an exaggeration, points out that attention isn't the only export from language processing into computer vision: tokenizing inputs, *i.e.*, using patch embeddings, is also a powerful and important takeaway.

While our model is not state-of-the-art, we find its simple patch-mixing design to be compelling. We hope that ConvMixers can serve as a baseline for future patch-based architectures with novel operations, or that they can provide a basic template for new conceptually simple and performant models.

Given that such simple architectures as ConvMixer can be successful, we question the role of continued architecture searches; in particular, are more complicated architectures fundamentally better at modeling phenomena, or are they ultimately just more computationally efficient? Much of the variance in accuracies may be explained by more advanced training pipelines and augmentation techniques, as demonstrated by Wightman et al. (2021) and our work.

**Future work.** We are optimistic that a deeper ConvMixer with larger patches could reach a desirable tradeoff between accuracy, parameters, and throughput after longer training and more regularization and hyperparameter tuning, similarly to how Wightman et al. (2021) enhanced ResNet performance through carefully-designed training regimens. Low-level optimization of large-kernel depthwise convolution could substantially increase throughput, and small enhancements to our architecture like the addition of bottlenecks or a more expressive classifier could trade simplicity for performance.

Due to its large internal resolution and isotropic design, ConvMixer may be especially well-suited for semantic segmentation, and it would be useful to run experiments on this task with a ConvMixer-like model and on other tasks such as object detection. More experiments could be designed to more clearly extricate the effect of patch embeddings from other architectural choices. In particular, for a more in-depth comparison to ViTs and MLP-Mixers, which excel when trained on very large data sets, it is important to investigate the performance of ConvMixers in the regime of large-scale pre-training.

More work is necessary to extricate the effect of the patch embeddings from the rest of the architecture. In particular, we have preliminary evidence that it is not necessary to separate the spatial and channel mixing steps; patches followed by any stack of nonlinear operations (say, plain convolution) may be sufficient for simple, performant models.

**Note on paper length.** We acknowledge that this paper is shorter than most, and this is intentional. In the main text, we have presented our main thesis, proposed an extremely simple architecture used to validate the thesis, included a complete implementation, highlighted the results that we believe to be most relevant, and finished with concluding thoughts. The work here is very simple, and thus we believe that a short paper is ultimately more effective at conveying the main messages. While additional experiments and results are included in the appendix, we fully argue that the results in the main text are sufficient to establish our point, and that the supplementary material is genuinely of secondary importance. Hence, we felt the shorter length was more than sufficient.

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

# A    Comparison to other models

Table 2: Throughputs measured on an RTX8000 GPU using batch size 64 and fp16. ConvMixers and ResNets trained ourselves. Other statistics: DeiT (Touvron et al., 2020), ResMLP (Touvron et al., 2021a), Swin (Liu et al., 2021b), ViT (Dosovitskiy et al., 2020), MLP-Mixer (Tolstikhin et al., 2021), Isotropic MobileNets (Sandler et al., 2019). We think models with matching colored dots (●) are informative to compare with each other. †Throughput tested, but not trained. Activations: **R**eLU, **G**ELU.
★Using new, better regularization hyperparameters based on Wightman et al. (2021)'s A1 procedure.

| Comparison with other simple models trained on **ImageNet-1k only** with input size 224. | | | | | | | |
|---|---|---|---|---|---|---|---|
| Network | Patch Size | Kernel Size | # Params ($\times 10^6$) | Throughput (img/sec) | Act. Fn. | # Epochs | ImNet top-1 (%) |
| ConvMixer-1536/20★ | 7 | 9 | 51.6 | 134 | G | 150 | 82.20 |
| ConvMixer-1536/20 ● | 7 | 9 | 51.6 | 134 | G | 150 | 81.37 |
| ConvMixer-1536/20★ | 7 | 3 | 49.4 | 246 | G | 150 | 81.60 |
| ConvMixer-1536/20 | 7 | 3 | 49.4 | 246 | G | 150 | 80.43 |
| ConvMixer-1536/20 | 14 | 9 | 52.3 | 538 | G | 150 | 78.92 |
| ConvMixer-1536/24★ | 14 | 9 | 62.3 | 447 | G | 150 | 80.21 |
| ConvMixer-768/32★ | 7 | 7 | 21.1 | 206 | G | 150 | 80.74 |
| ConvMixer-768/32 ● | 7 | 7 | 21.1 | 206 | R | 300 | 80.16 |
| ConvMixer-1024/16 | 7 | 9 | 19.4 | 244 | G | 100 | 79.45 |
| ConvMixer-1024/12 | 7 | 8 | 14.6 | 358 | G | 90 | 77.75 |
| ConvMixer-512/16 | 7 | 8 | 5.4 | 599 | G | 90 | 73.76 |
| ConvMixer-512/12 ● | 7 | 8 | 4.2 | 798 | G | 90 | 72.59 |
| ConvMixer-768/32 | 14 | 3 | 20.2 | 1235 | R | 300 | 74.93 |
| ConvMixer-1024/20 ● | 14 | 9 | 24.4 | 750 | G | 150 | 76.94 |
| ResNet-152★ | – | 3 | 60.2 | 828 | R | 150 | 81.15 |
| ResNet-152 ● | – | 3 | 60.2 | 828 | R | 150 | 79.64 |
| ResNet-101 ● | – | 3 | 44.6 | 1187 | R | 150 | 78.33 |
| ResNet-50 | – | 3 | 25.6 | 1739 | R | 150 | 76.32 |
| DeiT-B† | 7 | – | 86.7 | 83 | G | – | – |
| DeiT-S† | 7 | – | 22.1 | 174 | G | – | – |
| DeiT-Ti† | 7 | – | 5.7 | 336 | G | – | – |
| DeiT-B ● | 16 | – | 86 | 792 | G | 300 | 81.8 |
| DeiT-S ● | 16 | – | 22 | 1610 | G | 300 | 79.8 |
| DeiT-Ti ● | 16 | – | 5.7 | 2603 | G | 300 | 72.2 |
| ResMLP-S12/8 ● | 8 | – | 22.1 | 872 | G | 400 | 79.1 |
| ResMLP-B24/8 ● | 8 | – | 129 | 181 | G | 400 | 81.0 |
| ResMLP-B24 | 16 | – | 116 | 1597 | G | 400 | 81.0 |
| Swin-S ● | 4 | – | 50 | 576 | G | 300 | 83.0 |
| Swin-T ● | 4 | – | 29 | 878 | G | 300 | 81.3 |
| ViT-B/16 ● | 16 | – | 86 | 789 | G | 300 | 77.9 |
| Mixer-B/16 ● | 16 | – | 59 | 1025 | G | 300 | 76.44 |
| Isotropic MobileNetv3 ● | 8 | 3 | 20 | 355 | R | – | 80.6 |
| Isotropic MobileNetv3 ● | 16 | 3 | 20 | 1296 | R | – | 77.6 |

**Experiment overview.** We did not design our experiments to maximize accuracy: We chose "common sense" parameters for `timm` and its augmentation settings, found that it worked well for a ConvMixer-1024/12, and stuck with them for the proceeding experiments. We admit this is not an optimal strategy, however, we were aware from our early experiments on CIFAR-10 that results seemed robust to various small changes. We did not have access to sufficient compute to attempt to tune hyperparameters for each model: *e.g.*, larger ConvMixers could probably benefit from more regularization than we chose, and smaller ones from less regularization. Keeping the parameters the same across ConvMixer instances seemed more reasonable than guessing for each.

However, to some extent, we changed the number of epochs per model: for earlier experiments, we merely wanted a "proof of concept", and used only 90–100 epochs. Once we saw potential, we increased this to 150 epochs and trained some larger models, namely ConvMixer-1024/20 with $p = 14$ patches and ConvMixer-1536/20 with $p = 7$ patches. Then, believing that we should explore deeper-but-less-wide ConvMixers, and knowing from CIFAR-10 that the deeper models converged more slowly, we trained ConvMixer-768/32s with $p = 14$ and $p = 7$ for 300 epochs. Of course, training time was a consideration: ConvMixer-1536/20 took about 9 days to train (on $10\times$ RTX8000s) 150 epochs, and ConvMixer-768/32 is over twice as fast, making 300 epochs more feasible.

If anything, we believe that in the worst case, the lack of parameter tuning in our experiments resulted in underestimating the accuracies of ConvMixers. Further, due to our limited compute and the fact that large models (particularly ConvMixers) are expensive to train on large data sets, we generally trained our models for fewer epochs than competition like DeiT and ResMLP (see Table 2).

In this revision, we have added some additional results (denoted with a ★ in Table 2) using hyperparameters loosely based on the precisely-crafted "A1 training procedure" from Wightman et al. (2021). In particular, we adjusted parameters for RandAug, Mixup, CutMix, Random Erasing, and weight decay to match those in the procedure. Importantly, we still only trained for 150 epochs, rather than the 600 epochs used in Wightman et al. (2021), and we did not use binary cross-entropy loss nor repeated augmentation. While we do not think optimal hyperparameters for ResNet would also be optimal for ConvMixer, these settings are significantly better than the ones we initially chose. This further highlights the capabilities of ConvMixers, and we are optimistic that further tuning could lead to still-better performance. Throughout the paper, we still refer to ConvMixers trained using our initial "one shot" selection of hyperparameters.

**A note on throughput.** We measured throughput using batches of 64 images in half precision on a single RTX8000 GPU, averaged over 20 such batches. In particular, we measured CUDA execution time rather than "wall-clock" time. We noticed discrepancies in the relative throughputs of models, *e.g.*, Touvron et al. (2020) reports that ResNet-152 is $2\times$ faster than DeiT-B, but our measurements show that the two models have nearly the same throughput. We therefore speculate that our throughputs may underestimate the performance of ResNets and ConvMixers relative to the transformers. The difference may be due to using RTX8000 rather than V100 GPUs, or other low-level differences. Our throughputs were similar for batch sizes 32 and 128.

**ResNets.** As a simple baseline to which to compare ConvMixers, we trained three standard ResNets using exactly the same training setup and parameters as ConvMixer-1536/20. We also trained ResNet-152★ using the new A1-based procedure for comparison against ConvMixer-1536/20★. Despite having fewer parameters and being architecturally much simpler, ConvMixers substantially outperform these ResNets in terms of accuracy. A possible confounding factor is that ConvMixers use GELU, which may boost performance, while ResNets use ReLU. In an attempt to rule out this confound, we used ReLU in a later ConvMixer-768/32 experiment and found that it still achieved competitive accuracy. We also note that the choice of ReLU *vs.* GELU was not important on CIFAR-10 experiments (see Table 7). However, ConvMixers do have substantially less throughput.

**DeiTs.** We believe that DeiT is the most reasonable comparison in terms of vision transformers: It only adds additional regularization, as opposed to architectural additions in the case of CaiT (Touvron et al., 2021b), and is then essentially a "vanilla" ViT modulo the distillation token (we don't consider distilled architectures). In terms of a fixed parameter budget, ConvMixers generally outperform DeiTs. For example, ConvMixer-1536/20 is only 0.43% less accurate than DeiT-B despite having over 30M fewer parameters; ConvMixer-

768/32 is 0.36% more accurate than DeiT-S despite having 0.9M fewer parameters; and ConvMixer-512/16 is 0.39% more accurate than DeiT-Ti for nearly the same number of parameters. Admittedly, none of the ConvMixers are very competitive in terms of throughput, with the closest being the ConvMixer-512/16 which is 4× slower than DeiT-Ti.

A confounding factor is the difference in patch size between DeiT and ConvMixer; DeiT uses $p = 16$ while ConvMixer uses $p = 7$. This means DeiT is substantially faster. However, ConvMixers using larger patches are not as competitive. While we were not able to train DeiTs with larger patch sizes, it is possible that they would outperform ConvMixers on the parameter count *vs.* accuracy curve; however, we tested their throughput for $p = 7$, and they are even slower than ConvMixers. Given the difference between convolution and self-attention, we are not sure it is salient to control for patch size differences.

DeiTs were subject to more hyperparameter tuning than ConvMixers, as well as longer training times. They also used stochastic depth while we did not, which can in some cases contribute percent differences in model accuracy (Touvron et al., 2021a). It is therefore possible that further hyperparameter tuning and more epochs for ConvMixers could close the gap between the two architectures for large patches, *e.g.*, $p = 16$.

**ResMLPs.** Similarly to DeiT for ViT, we believe that ResMLP is the most relevant MLP-Mixer variant to compare against. Unlike DeiT, we can compare against instances of ResMLP with similar patch size: ResMLP-B24/8 has $p = 8$ patches, and underperforms ConvMixer-1536/20 by 0.37%, despite having over twice the number of parameters; it also has similarly low throughput. ConvMixer-768/32 also outperforms ResMLP-S12/8 for millions fewer parameters, but 4× less throughput.

ResMLP did not significantly improve in terms of accuracy for halving the patch size from 16 to 8, which shows that smaller patches do not always lead to better accuracy for a fixed architecture and regularization strategy (*e.g.*, training a $p = 8$ DeiT may be challenging).

**Swin Transformers.** While we intend to focus on the most basic isotropic, patch-based architectures for fair comparisons with ConvMixer, it is also interesting to compare to a more complicated model that is closer to state-of-the-art. For a similar parameter budget, ConvMixer is around 1.2-1.6% less accurate than the Swin Transformer, while also being 4-6× slower. However, considering we did not attempt to tune or optimize our model in any way, we find it surprising that an exceedingly simple patch-based model that uses only plain convolution does not lag too far behind Swin Transformer.

**Isotropic MobileNets.** These models are closest in design to ours, despite using a repeating block that is substantially more complex than the ConvMixer one. Despite this, for a similar number of parameters, we can get similar performance. Notably, isotropic MobileNets seem to suffer less from larger patch sizes than ConvMixers, which makes us optimistic that sufficient parameter tuning could lead to more performant large-patch ConvMixers. As Sandler et al. (2019) did not provide an implementation, we cannot be sure if ours is exactly the same; *e.g.*, we were unsure if 5x5 stride-5 convolutions were replaced with 3x3 or 5x5 stride-1 convolutions, so we chose 3x3. The throughputs in Table 2 are based on our implementation. We also trained a patch-size-16 Isotropic MobileNet using exactly the same pipeline used for our ConvMixers, which achieved only 70.76% accuracy.

**Other models.** We included ViT and MLP-Mixer instances in our table, though they are not competitive with ConvMixer, DeiT, or ResMLP, even though MLP-Mixer has comparable regularization to ConvMixer. That is, ConvMixer seems to outperform MLP-Mixer and ViT, while being closer to complexity to them in terms of design and training regime than the other competitors, DeiT and ResMLP.

**Kernel size.** While we found some evidence that larger kernels are better on CIFAR-10, we wanted to see if this finding transferred to ImageNet. Consequently, we trained our best-performing model, ConvMixer-1536/20, with kernel size $k = 3$ rather than $k = 9$. This resulted in a decrease of 0.94% top-1 accuracy, which we believe is quite significant relative to the mere 2.2M additional parameters. However, $k = 3$ is substantially faster than $k = 9$ for spatial-domain convolution; we speculate that low-level optimizations could close the performance gap to some extent, *e.g.*, by using implicit instead of explicit padding. Since large-kernel convolutions throughout a model are unconventional, there has likely been low demand for such optimizations.

# B  Additional Experiments on ImageNet

In this section, we present additional experiments on ImageNet-1k. We primarily used ConvMixer-512/12 trained using the new A1-like (★) technique. Note that the throughputs in this section were recorded using Tesla V100 GPUs, while those in Table 2 used RTX8000s (hence, the two measurements should not be compared across tables).

Table 3: We investigate the effect of different patch sizes on throughput and accuracy. Smaller patches result in higher accuracy at the expense of throughput.

| Effect of Patch Size | | | | |
|---|---|---|---|---|
| Network | Patch Size | Kernel Size | Throughput (img/sec) | ImNet top-1 (%) |
| ConvMixer-512/12 | 5 | 9 | 388 | 75.60 |
| ConvMixer-512/12 | 7 | 9 | 644 | 74.60 |
| ConvMixer-512/12 | 9 | 9 | 1120 | 73.55 |
| ConvMixer-512/12 | 12 | 9 | 1908 | 71.79 |
| ConvMixer-512/12 | 16 | 9 | 2892 | 69.65 |

**Patch sizes.** Larger patch sizes result in lower accuracy, while smaller patches increase accuracy. However, ConvMixers using smaller patches are substantially slower. For most of our experiments, we used $7 \times 7$ patches; however, in some cases, it may be desirable to use slightly larger $9 \times 9$ patches in exchange for a bit less accuracy (see Table 3).

Table 4: We tested ConvMixers with ResNet-style stems and ResNets with patch embedding stems; in both cases, patch embeddings worked better.

| Patch Embeddings vs. ResNet-Style Stems | | |
|---|---|---|
| Network | Stem | ImNet top-1 (%) |
| ResNet50 | ResNet Stem | 78.32 |
| ResNet50 | Patches ($4 \times 4$) | **78.74** |
| ConvMixer-512/12 | ResNet Stem | 71.24 |
| ConvMixer-512/12 | Patches ($12 \times 12$) | **71.79** |

**Disentangling the effect of patches.** We found that using a patch embedding stem with a ResNet improves accuracy relative to the default stem, while using a ResNet stem with a ConvMixer hurts accuracy (see Table 4). This provides some evidence that patches are a good choice of input representation, and may even improve the performance of existing models compared to their default input representation. For the ConvMixer, we used a ResNet stem with $12 \times 12$-kernel convolutions with stride 6 followed by max pooling; this ensured that ResNet-stem ConvMixer had the same internal resolution as the version using patches.

**Kernel sizes.** Here, we investigate whether larger kernel sizes are really beneficial to ConvMixers. In Table 5, we see that $9 \times 9$ kernels strongly outperform $3 \times 3$ kernels. This may be unsurprising, as the model with $9 \times 9$ kernels has significantly more parameters; to control for this, we trained a ConvMixer-512/14 with $3 \times 3$ kernels which has a comparable number of parameters. However, this still does not achieve the performance of the $9 \times 9$-kernel model. Further, conventional wisdom states that three stacked $3 \times 3$ convolutional layers (with GELUs between the layers) has the same receptive field as $9 \times 9$ convolution while being more expressive. Consequently, we replaced plain $3 \times 3$ convolutions with three stacked $3\times$ convolutions; however, this still did not surpass the accuracy of $9 \times 9$ convolutions. Finally, using the same intuition, we stack three ConvMixer-512/12s and tried a ConvMixer-512/36; only then do we outperform large-kernel convolutions. This is perhaps unsurprising, given the 24 additional pointwise layers.

Table 5: Here, we investigate whether larger kernels are really more effective than smaller ones. Our results suggest that larger kernels are advantageous compared to a variety of "control" experiments.

| Effect of Kernel Size | | | | | |
|---|---|---|---|---|---|
| Network | Patch Size | Kernel Size | # Params ($\times 10^6$) | Throughput (img/sec) | ImNet top-1 (%) |
| ConvMixer-512/12 | 7 | 7 | 4.07 | 724 | 74.54 |
| ConvMixer-512/12 | 7 | 15 | 5.15 | 401 | 75.25 |
| ConvMixer-512/12 | 7 | 9 | 4.27 | 644 | 74.60 |
| ConvMixer-512/12 | 7 | 3 | 3.83 | 992 | 72.96 |
| ConvMixer-512/14 | 7 | 3 | 4.37 | 856 | 74.03 |
| ConvMixer-512/12 (3 stacked 3x3-GELU convs) | 7 | 3 | 3.95 | 732 | 74.53 |
| ConvMixer-512/36 | 7 | 3 | 10.3 | 338 | 77.67 |

Table 6: We investigated choices of activation functions and normalization layers, as well as training with reduced data augmentation. While reducing augmentation improves performance on this small model, we did not adopt this change elsewhere.

| Ablation of ConvMixer-512/12 on ImageNet | |
|---|---|
| Ablation | ImNet Acc. (%) |
| Baseline | 74.60 |
| BatchNorm → LayerNorm | 74.51 |
| GELU → ReLU | 74.44 |
| – Mixup and CutMix | 75.65 |
| – RandAug | 75.26 |

**Architectural choices.** In Table 6, we demonstrate that the choice of activation function (ReLU vs. GELU) and norm layer (BatchNorm vs. LayerNorm) does not have a large impact on performance.

**Data augmentation.** We also investigate removing some of the data augmentations from the A1 recipe (see Table 6). We saw a substantial performance boost from removing Mixup and CutMix, and to a lesser extent, RandAugment as well. This is likely due to the relatively small model used for the comparison (ConvMixer-512/12), for which this level of augmentation may be excessive. We did not adopt these changes for other experiments. For comparison, a DeiT trained exactly the same way as the baseline ConvMixer achieves 70.28% accuracy, while a DeiT without RandAug, CutMix, and MixUp gets 69.65% accuracy. That is, it seems augmentations are more important to DeiT than to ConvMixer.

**Input size.** Unlike ViTs, MLP-Mixers, ResMLPs, and other recent models, ConvMixers can handle variable input sizes with no modifications whatsoever. In Fig. 4, we show the effect of input size on the inference time of a ConvMixer-768/32 using a batch size of 32, averaged over 16 trials on an RTX 3080Ti GPU in half precision. Note the rapid growth of inference time for kernel sizes 7 and 9 compared to 3 and 5; we believe this shows that the underlying implementation of depthwise convolution is suboptimal for large kernel sizes.

## C  Additional Experiments on CIFAR-10

**Residual connections.** We experimented with leaving out one, the other, or both residual connections before settling on the current configuration, and consequently chose to leave out the second residual connection. Our baseline model without the connection achieves 95.88% accuracy, while including the connection

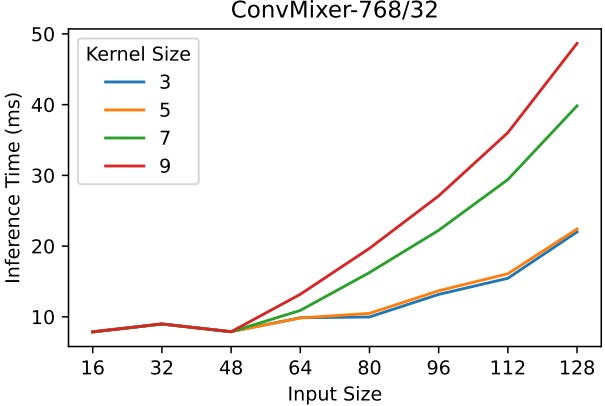

Figure 4: Inference time vs. input size for ConvMixer-768/32 with a variety of kernel sizes.

reduces it to 94.78%. Surprisingly, we see only a 0.31% decrease in accuracy for *removing all residual connections*. We acknowledge that these findings for residual connections may not generalize to deeper ConvMixers trained on larger data sets.

Table 7: Small ablation study of training a ConvMixer-256/8 on CIFAR-10.

| Ablation of ConvMixer-256/8 on CIFAR-10 | |
|---|---|
| Ablation | CIFAR-10 Acc. (%) |
| Baseline | 95.88 |
| – Residual in Eq. 2 | 95.57 |
| + Residual in Eq. 3 | 94.78 |
| BatchNorm → LayerNorm | 94.44 |
| GELU → ReLU | 95.51 |
| – Mixup and CutMix | 95.92 |
| – Random Erasing | 95.24 |
| – RandAug | 92.86 |
| – Random Scaling | 86.24 |
| – Gradient Norm Clipping | 86.33 |

**Normalization.** Our model is conceptually similar to the vision transformer and MLP-Mixer, both of which use LayerNorm instead of BatchNorm. We attempted to use LayerNorm instead, and saw a decrease in performance of around 1% as well as slower convergence (see Table 7). However, this was for a relatively shallow model, and we cannot guarantee that LayerNorm would not hinder ImageNet-scale models to an even larger degree. We note that the authors of ResMLP also saw a relatively small increase in accuracy for replacing LayerNorm with BatchNorm, but for a larger-scale experiment (Touvron et al., 2021a). We conclude that BatchNorm is no more crucial to our architecture than other regularizations or parameter settings (*e.g.*, kernel size).

Having settled on an architecture, we proceeded to adjust its parameters $h, d, p, k$ as well as weight decay on CIFAR-10 experiments. (Initially, we took the unconventional approach of excluding weight decay since we were already using strong regularization in the form of RandAug and mixup.) We acknowledge that tuning our architecture on CIFAR-10 does not necessarily generalize to performance on larger data sets, and that this is a limitation of our study.

### C.1 Results

ConvMixers are quite performant on CIFAR-10, easily achieving $> 91\%$ accuracy for as little as $100,000$ parameters, or $> 96\%$ accuracy for only $887,000$ parameters (see Table 8). With additional refinements *e.g.*, a more expressive classifier or bottlenecks, we think that ConvMixer could be even more competitive. For all experiments, we trained for 200 epochs on CIFAR-10 with RandAug, mixup, cutmix, random erasing, gradient norm clipping, and the standard augmentations in `timm`. We remove some of these augmentations in Table 7, finding that RandAug and random scaling ("default" in `timm`) are very important, each accounting for over 3% of the accuracy.

**Scaling ConvMixer.** We adjusted the hidden dimension $h$ and the depth $d$, finding that deeper networks take longer to converge while wider networks converge faster. That said, increasing the width or the depth is an effective way to increase accuracy; a doubling of depth incurs less compute than a doubling of width. The number of parameters in a ConvMixer is given exactly by:

$$\#\mathsf{params} = h[d(k^2 + h + 6) + c_{\mathsf{in}}p^2 + n_{\mathsf{classes}} + 3] + n_{\mathsf{classes}}, \tag{4}$$

including affine scaling parameters in BatchNorm layers, convolutional kernels, and the classifier.

**Kernel size.** We initially hypothesized that large kernels would be important for ConvMixers, as they would allow the mixing of distant spatial information similarly to unconstrained MLPs or self-attention layers. We tried to investigate the effect of kernel size on CIFAR-10: we fixed the model to be a ConvMixer-256/8, and increased the kernel size by 2s from 3 to 15.

Using a kernel size of 3, the ConvMixer only achieves 93.61% accuracy. Simply increasing it to 5 gives an additional 1.50% accuracy, and further to 7 an additional 0.61%. The gains afterwards are relatively marginal, with kernel size 15 giving an additional 0.28% accuracy. It could be that with more training iterations or more regularization, the effect of larger kernels would be more pronounced. Nonetheless, we concluded that ConvMixers benefit from larger-than-usual kernels, and thus used kernel sizes 7 or 9 in most of our later experiments.

It is conventional wisdom that large-kernel convolutions can be "decomposed" into stacked small-kernel convolutions with activations between them, and it is therefore standard practice to use $k = 3$ convolutions, stacking more of them to increase the receptive field size with additional benefits from nonlinearities. This raises a question: is the benefit of larger kernels in ConvMixer actually better than simply increasing the depth with small kernels? First, we note that deeper networks are generally harder to train, so by increasing the kernel size independently of the depth, we may recover some of the benefits of depth without making it harder for signals to "propagate back" through the network. To test this, we trained a ConvMixer-256/10 with $k = 3$ (698K parameters) in the same setting as a ConvMixer-256/8 with $k = 9$ (707K parameters), *i.e.*, we increased depth in a small-kernel model to roughly match the parameters of a large-kernel model. The ConvMixer-256/10 achieved 94.29% accuracy (1.5% less), which provides more evidence for the importance of larger kernels in ConvMixers. Next, instead of fixing the parameter budget, we tripled the depth (using the intuition that 3 stacked $k = 3$ convolutions have the receptive field of a $k = 9$ convolution), giving a ConvMixer-256/24 with 1670K parameters, and got 95.16% accuracy, *i.e.*, still less.

**Patch size.** CIFAR-10 inputs are so small that we initially only used $p = 1$, *i.e.*, the patch embedding layer does little more than compute $h$ linear combinations of the input image. Using $p = 2$, we see a reduction in accuracy of about 0.80%; this is a worthy tradeoff in terms of training and inference time. Further increasing the patch size leads to rapid decreases in accuracy, with only 92.61% for $p = 4$.

Since the "internal resolution" is decreased by a factor of $p$ when increasing the patch size, we assumed that larger kernels would be less important for larger $p$. We investigated this by again increasing the kernel size from 3 to 11 for ConvMixer-256/8 with $p = 2$: however, this time, the improvement going from 3 to 5 is only 1.13%, and larger kernels than 5 provide only marginal benefit.

**Weight decay.** We did many of our initial experiments with minimal weight decay. However, this was not optimal: by tuning weight decay, we can get an additional 0.15% of accuracy for no cost. Consequently, we used weight decay (without tuning) for our larger-scale experiments on ImageNet.

Table 8: An investigation of ConvMixer design parameters $h, d, p, k$ and weight decay on CIFAR-10

| Width $h$ | Depth $d$ | Patch Size $p$ | Kernel Size $k$ | # Params ($\times 10^3$) | Weight Decay | CIFAR-10 Acc. (%) |
|---|---|---|---|---|---|---|
| Tiny **ConvMixers** trained on CIFAR-10. | | | | | | |
| 128 | 4 | 1 | 8 | 103 | 0 | 91.26 |
| 128 | 8 | 1 | 8 | 205 | 0 | 93.83 |
| 128 | 12 | 1 | 8 | 306 | 0 | 94.83 |
| 256 | 4 | 1 | 8 | 338 | 0 | 93.37 |
| 256 | 8 | 1 | 8 | 672 | 0 | 95.60 |
| 256 | 12 | 1 | 8 | 1006 | 0 | 96.39 |
| 256 | 16 | 1 | 8 | 1339 | 0 | 96.74 |
| 256 | 20 | 1 | 8 | 1673 | 0 | 96.67 |
| ↓ Kernel adjustments | | | | | | |
| 256 | 8 | 1 | 3 | 559 | 0 | 93.61 |
| 256 | 8 | 1 | 5 | 592 | 0 | 95.19 |
| 256 | 8 | 1 | 7 | 641 | 0 | 95.80 |
| 256 | 8 | 1 | 9 | 707 | 0 | 95.88 |
| 256 | 8 | 1 | 11 | 788 | 0 | 95.70 |
| 256 | 8 | 1 | 13 | 887 | 0 | 96.04 |
| 256 | 8 | 1 | 15 | 1001 | 0 | 96.08 |
| ↓ Patch adjustments | | | | | | |
| 256 | 8 | 2 | 9 | 709 | 0 | 95.00 |
| 256 | 8 | 4 | 9 | 718 | 0 | 92.61 |
| 256 | 8 | 8 | 9 | 755 | 0 | 85.57 |
| ↓ Weight decay adjustments | | | | | | |
| 256 | 8 | 1 | 9 | 707 | $1 \times 10^{-1}$ | 95.88 |
| 256 | 8 | 1 | 9 | 707 | $1 \times 10^{-2}$ | 96.03 |
| 256 | 8 | 1 | 9 | 707 | $1 \times 10^{-3}$ | 95.76 |
| 256 | 8 | 1 | 9 | 707 | $1 \times 10^{-4}$ | 95.63 |
| 256 | 8 | 1 | 9 | 707 | $1 \times 10^{-5}$ | 95.88 |
| ↓ Kernel size adjustments when $p = 2$ | | | | | | |
| 256 | 8 | 2 | 3 | 561 | 0 | 94.08 |
| 256 | 8 | 2 | 5 | 594 | 0 | 95.21 |
| 256 | 8 | 2 | 7 | 643 | 0 | 95.35 |
| 256 | 8 | 2 | 9 | 709 | 0 | 95.00 |
| 256 | 8 | 2 | 11 | 791 | 0 | 95.14 |
| ↓ Adding weight decay to the above | | | | | | |
| 256 | 8 | 2 | 3 | 561 | $1 \times 10^{-2}$ | 94.69 |
| 256 | 8 | 2 | 5 | 594 | $1 \times 10^{-2}$ | 95.26 |
| 256 | 8 | 2 | 7 | 643 | $1 \times 10^{-2}$ | 95.25 |
| 256 | 8 | 2 | 9 | 709 | $1 \times 10^{-2}$ | 95.06 |
| 256 | 8 | 2 | 11 | 791 | $1 \times 10^{-2}$ | 95.17 |

# D  Weight Visualizations

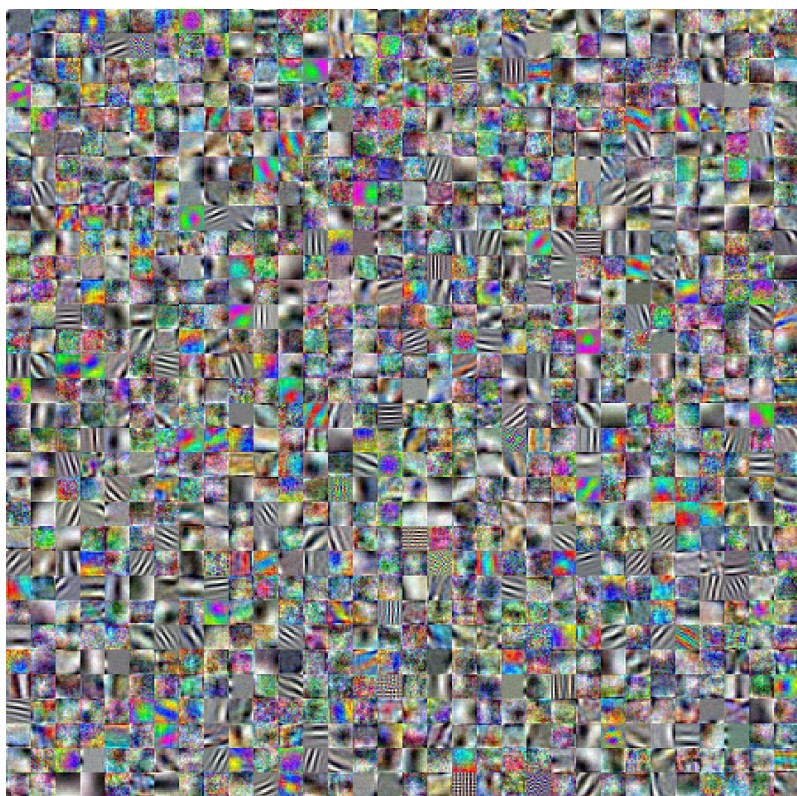

Figure 5: Patch embedding weights for a ConvMixer-1024/20 with patch size 14 (see Table 2).

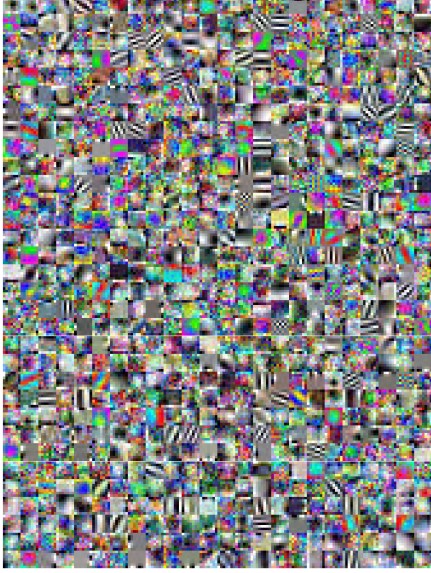

Figure 6: Patch embedding weights for a ConvMixer-768/32 with patch size 7 (see Table 2).

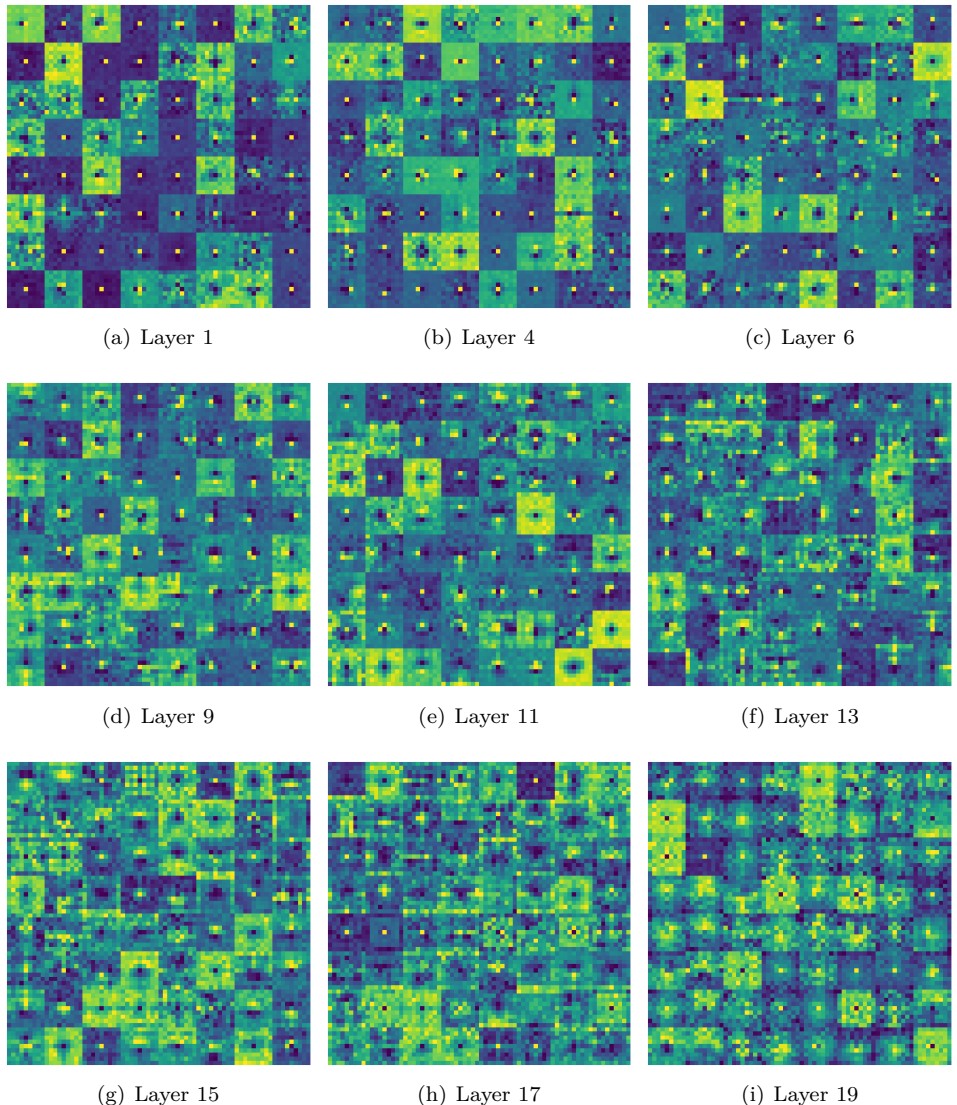

Figure 7: Random subsets of 64 depthwise convolutional kernels from progressively deeper layers of ConvMixer-1536/20 (Table 1).

In Figure 5 and 6, we visualize the (complete) weights of the patch embedding layers of a ConvMixer-1536/20 with $p = 14$ and a ConvMixer-768/32 with $p = 7$, respectively. Much like Sandler et al. (2019), the layer consists of Gabor-like filters as well as "colorful globs" or rough edge detectors. The filters seem to be more structured than those learned by MLP-Mixer (Tolstikhin et al., 2021); also unlike MLP-Mixer, the weights look much the same going from $p = 14$ to $p = 7$: the latter simply looks like a downsampled version of the former. It is unclear, then, why we see such a drop in accuracy for larger patches. However, some of the filters essentially look like noise, maybe suggesting a need for more regularization or longer training, or even more data. Ultimately, we cannot read too much into the learned representations here.

In Figure 7, we plot the hidden convolutional kernels for successive layers of a ConvMixer. Initially, the kernels seem to be relatively small, but make use of their allowed full size in later layers; there is a clear hierarchy of features as one would expect from a standard convolutional architecture. Interestingly, Touvron et al. (2021a) saw a similar effect for ResMLP, where earlier layers look like small-kernel convolution, while later layers were more diffuse, despite these layers being represented by an unconstrained matrix multiplication rather than convolution.

## E Implementation

```
1  def ConvMixer(h,d,k,p,n):
2   S,C,A=Sequential,Conv2d,lambda x:S(x,GELU(),BatchNorm2d(h))
3   R=type('',(S,),{'forward':lambda s,x:s[0](x)+x})
4   return S(A(C(3,h,p,p)),*[S(R(A(C(h,h,k,groups=h,padding=k//2))),A(C(h,h,1))) for i in range(d)],
5       AdaptiveAvgPool2d(1),Flatten(),Linear(h,n))
```

Figure 8: An implementation of our model in less than 280 characters, in case you happen to know of any means of disseminating information that could benefit from such a length.

All you need to do to run this is `from torch.nn import *`.

We present an *even more terse* implementation of ConvMixer in Figure 8, which to the best of our knowledge is the first model that achieves the elusive dual goals of 82%+ ImageNet top-1 accuracy while also fitting into a tweet.

