# OpenReview forum: "Patches Are All You Need?"
_TMLR — Accepted by TMLR_

### Review · Reviewer_7eQ2 · 2023-01-31

**Summary Of Contributions:**

The authors consider the success of Vision Transformers and other recent computer vision models such as MLP-Mixer. The authors devise an architecture with similar attributes: the patch embedding and an isotropic architecture with alternating spatial and channel dimension mixing, but they use convolutions instead of attention or MLPs from prior work. The resulting architecture, ConvMixer, is simple compared to prior work, both conceptually and in terms of the number of parameters, yet the authors show it achieves fairly strong performance on ImageNet even without many optimizations.

The main contributions are as follows
 - They release a new baseline that is conceptually simple and lightweight, and achieves good performance, allowing for ease of follow-up work.
 - They give evidence that patches+convolutions are sufficient for designing high-performing vision models. This acts as an ablation study for prior works that used attention or MLPs for vision.

**Audience:**

Yes

**Broader Impact Concerns:**

There are no broader impact concerns.

**Claims And Evidence:**

Yes

**Requested Changes:**

None, but I encourage the authors to address the points in `weaknesses' above, which would make this paper even stronger.

**Strengths And Weaknesses:**

## Strengths

- Simplicity. Simple approaches are more practical and applicable than complex approaches. This will be an important baseline moving forward.
- An ablation for the works that use transformers/MLPs for vision, showing that patches may be the critical component.
- The paper is well-written, clear, and honest.

Due to these strengths, I believe this paper will have high impact.

## Weaknesses

1. **Simplicity of MLPs vs convolutions.** The insight, “patches are all we need”, already has partial evidence from two prior works: MLP-Mixer and Melas-Kyriazi (2021). These works replaced the attention layers in the original Vision Transformer with simple components, namely MLPs. In fact, MLPs are conceptually simpler than convolutions (although, the authors argue that convolutions are less expressive than MLPs).
Similarly, the authors use the phrase “new operations (like self-attention and MLPs)”  but MLPs are not newer than convolutions.
That said, ConvMixers do still fill an important niche: they are significantly simpler than MLP-Mixer (which was optimized much more for performance), and significantly better-performing than Melas-Kyriazi -- 74.9% vs. 82.2% on ImageNet. This niche is important because it makes ConvMixer a strong baseline and easy to build off of.

2. **"Accuracies we report likely underestimate capabilities."**
I understand that the authors explicitly did not optimize their architecture for accuracy or inference speed, mentioning this several times throughout the text, because it was not necessary for their message, and because it costs a lot to train ImageNet models. I agree that this should not be necessary. The authors also compared to the most basic isotropic patch-based architectures (DeiT and ResMLP).
It is still somewhat of a downside, because (a) there is no way to tell how much the model will improve with HPO, and (b) there is no way to tell how much hyperparameter tuning has already been done by the authors (although they claim only light HPO). One option would be to compare ConvMixer with other approaches, each with HPO, on one of the popular versions of ImageNet that is a small subset and heavily downsampled.

3. **Table 2.** It is hard to interpret Table 2. Each training run has several decisions: width, depth, patch size, kernel size, activation function, and #epochs, and it is hard to tell why these 14 runs were chosen, and how they are ordered. The colored dots make it easier to see which results to compare, but that also begs the question, what is the point of the rows without colored dots?
Perhaps, whenever possible, specific experiments/insights can be shown in their own table (for example, Table 3 is a clean insight). The authors could also consider making Table 3 and similar experiments where just one variable is varied, into a plot.

I believe that the strengths clearly outweigh the weaknesses.

## Minor comment

**Paper length.**
The fact that the authors only use 6 pages is in stark contrast to the unspoken rules of the ML community (especially the NeurIPS, ICML, communities) where a paper must exactly hit the page limit and generally minimize whitespace, or else face a likely rejection. (I tried to find a citation for this beyond my own experience, and I could only find [this StackExchange thread](https://academia.stackexchange.com/questions/25650/is-it-a-good-idea-to-submit-paper-which-is-shorter-than-conferences-page-limit)).
While TMLR has a 12-page limit, generally papers below 9 pages (the NeurIPS limit) are seen as short.

However, papers should be reviewed based on their scientific merit, not their page count.

I agree with the authors that 6 pages is a good length for their paper. It is healthy for the community to remember to focus on scientific merit rather than reaching the page limit. Furthermore, too many “unspoken rules” make it harder for newcomers to join the ML community, especially rules that have to do with maximizing page numbers (e.g., there is a [new ICLR initiative](https://iclr.cc/Conferences/2023/CallForTinyPapers) to address this).

---

> ### Author Response · Authors · 2023-02-16
> **Response**
>
> Weaknesses
>
> 1. When we said “new operations”, we meant in the context of performant vision models – so while MLPs are themselves not new, reasonable accuracy on ImageNet with just MLPs is new. We agree it’s also rather subjective whether MLPs or convolutions are “simpler”. For one, convolutions are linear, while MLPs are not. And depending on the particular shape of the MLP, it may be able to implement, e.g., two convolutional layers separated by a nonlinearity. One point that may go against our claim is that convolutions have an inductive bias that is useful for vision tasks, while MLPs (presumably) do not. However, it is hard to explicitly measure such an inductive bias and weigh it against another rather vague term (simplicity). Either way, we agree with your assessment that ConvMixer is a strong and useful baseline for future comparisons and work.
>
> 2. We really, truly did no hyperparameter optimization on ImageNet – as shown in the appendix, we did some tuning on CIFAR-10, but presumably good parameters for CIFAR-10 are not necessarily ideal for ImageNet. The parameters we used are inspired by the work on CIFAR-10, the defaults for the timm library, and guessing (though this guess is the same across experiments unless otherwise noted). Later, we used parameters from “ResNet Strikes Back” [1] which were better than our initial guess. We thought this could be a point in our favor, in that if our model outperforms others without HPO, this establishes a sort of lower bound on ConvMixer’s advantage.
>
>    But you’re definitely right: it would be better if we could spend a fixed budget doing HPO for a variety of models for a somewhat more realistic comparison. But in practice, we simply don’t have the compute for this. Even just training some of the larger ConvMixers presented in the paper (once!) was quite difficult and expensive for us.
>
> 3. If Table 2 looks somewhat ad-hoc, that’s because it really is the result of an ad-hoc exploration. We wanted to find out what width, depth, kernel size, patch size, training time, and activation function made sense for our architecture, but did not have the compute to do this systematically. We started out with some smaller ConvMixers trained for fewer epochs, and worked our way up to larger ones. For example, we were curious if we could surpass 80% top-1 accuracy on ImageNet using 14x14 patches rather than 7x7, and ConvMixer-1536/24 was our best guess at a configuration that might work, based on previous runs (and it did…). It may be worth noting that this table plus the rest of the paper contain results for every ConvMixer we have ever trained on ImageNet (i.e., there were no “trial runs” – the “trial runs” are the results). At least part of the reason this table is in the appendix rather than the main paper is precisely because the exploration was so unstructured. (But we do still fully argue that the main paper makes our point -- we are just explaining that we very much viewed Table 2 as material best suited for the appendix.)
>
>     As for the table -> plot suggestion, we will look into that. We feel (not too strongly) that there are perhaps too few data points to make a plot that is substantially more useful than a table.
>
> Thanks for your discussion of paper length. We think the ICLR Tiny Papers initiative is a great idea and look forward to seeing how the idea evolves. We (of course) think that every conference-level scientific contribution does not necessarily require an N-page paper, especially in a field that progresses as quickly as ML.
>
> [1] https://arxiv.org/pdf/2110.00476.pdf

---

### Review · Reviewer_p1Lh · 2023-02-02

**Summary Of Contributions:**

The paper proposes a new architecture for image classification (and possible useful in other CV tasks) called ConvMixer which can be seen as a convolutional version of MLPMixer. Authors find that it is more parameter-efficient than ResNet, ResMLP and DeiT on ImageNet-1k and attribute the improvement to the use of patch representation as opposed to the more expressive attention / MLP transformations. The authors also analyze the effect of model hyperparameters (such as the patch size) to on the throughput and classification accuracy.

**Audience:**

Yes

**Broader Impact Concerns:**

No concerns.

**Claims And Evidence:**

Yes

**Requested Changes:**

I would be interested in hearing (reading) authors' opinion on what does it mean for us if indeed patches are all we need? Fundamentally,  I see no reason why this kind of representations should be globally optimal even withing the current knowledge frontier of deep learning. What do we learn beyond just this fact that would matter for finding more efficient architectures in the future?

**Strengths And Weaknesses:**

### Strengths

The paper investigates a single, simple and meaningful goal of understanding the effect of using patches and finding most (more?) efficient ways of processing them. I'm very curious to see if indeed ConvMixer will prove to be useful in other tasks, perhaps involving image-to-image transformations. I find the experiments done, again, in a clear and convincing manner.

### Weaknesses

This should not necessarily reflect negatively on the paper, but unlike in ViT and MLPMixer papers we do not necessarily understand better which are better universal building blocks for composing efficient architectures. Not all data modalities have analogs to patches and not every one that do will benefit from ConvMixer or at least that's what I expect. I still think that the paper findings are important and should not be underestimated even if my argument here is true.

---

> ### Author Response · Authors · 2023-02-16
> **Response**
>
> Thanks for your review!
>
> #### Weaknesses
>
> With respect to introducing convolutions to patch-based architectures, our takeaway was not necessarily that convolution is a “better” universal building block for architectures than self-attention or MLPs, but rather that a wide variety of operations (including convolution) work well within patch-based architectures. We would have to do a much more involved and computationally expensive study to understand which is the best “universal” operation (if such a task is even possible). It’s certainly true that not all modalities can be decomposed into patches, but our intention was for this to be a computer vision paper. That said, convolution can, in some sense, be applied to any data that has some notion of order in time or space. It may be the case that convolutional models will eventually perform as well as attention-based models, even on (maybe just some) language modeling tasks (but we of course don’t really know at this point) [1, 2].
>
> [1] https://arxiv.org/abs/2210.09298
>
> [2] https://arxiv.org/abs/2008.02496
>
>
> #### Requested changes
>
> We weren’t trying to argue that the patch-based representation is universally optimal in any sense. Rather, we were trying to argue that it is sufficient, and that it also makes the process of architecture design much simpler. For example: it removes a variable from the search, namely where to include downsampling/pooling layers, and it also aids in the decomposition of operations into spatial and channel mixing steps (conceptually, in our opinion – just because the hidden dimension is fixed after this layer).
>
> We think a better question related to our paper is as follows: what does it mean if more complicated architectures are basically just optimization tricks (in terms of speed/memory), and actual performance/representational ability can be captured just as effectively by extremely simple models? We suspect that hardware has co-evolved with software to some extent (or vice versa), insofar as architectures are popular if they achieve good accuracy while also being tailored to run fast on current GPUs (e.g., ResNets, MobileNets). Large (depthwise) convolutional kernels aren’t popular at least in part because they are excessively slow on current GPUs, though there is no fundamental reason for this to be the case. At this point, we have fairly strong evidence that larger kernels are better, but it’s unclear if the tradeoff makes sense in terms of, say, training time (with existing hardware and convolution implementations). Essentially, we wonder about decoupling speed from accuracy, and building models that are the best (in terms of accuracy) without regard for speed and then waiting for the hardware/low-level implementations to catch up. To be clear, we are just posing a question here; for the time being, efficient architectures are certainly very important (especially in the context of Transformer-style models!).

---

### Review · Reviewer_9PYK · 2023-02-07

**Summary Of Contributions:**

The authors propose a new model for computer vision tasks, named ConvMixer.
This model replaces modules within layers of a Vision Transformer (ViT) with fully convolutional operations:
self-attention is replaced with depthwise convolutions and MLPs with pointwise (1$\times$1) convolutions.
The design motivation stems from a simple question posed by the authors:
whether the success of ViT-style models could be due to the use of patches,
which are common components in such models.
The proposed ConvMixer is simple and parameter-efficient, and experiments show that it outperforms prevalent computer vision models
like ViT, MLP-Mixer, and classical ResNets, under controlled comparisons with similar datasets and model sizes.

**Audience:**

Yes

**Broader Impact Concerns:**

The impact of this work is no different than hundreds of existing computer vision models that are benchmarked on the ImageNet dataset.
It presents a simple scientific question and provides experimental evidence to support it.
Further discussion is not necessary for this context.


**Claims And Evidence:**

Yes

**Requested Changes:**

I would suggest changes according to Weaknesses (1, 2, 3) as per the authors' capacity and available computational resources.
In my opinion, none of them are critical to the core scientific merits of the paper,
but they would be valuable additions nonetheless for maximizing the potential impact of this paper.


**Strengths And Weaknesses:**

### Strengths:

This paper has a number of notable strengths; below I list a few salient strengths with supporting evidence:

1. **Presents food-for-thought to the vision community:**
   This paper presents poses a simple question that sounds very contrarian to the design philosophy of recent models:
   is the success of recent architectures like ViT and MLP-Mixer largely due to the early processing of images as patches?
   Vision Transformers were introduced with a motivation to bring the benefits of transformer architectures
   that have revolutionized the field of natural language processing (dubbed the "BERT moment").
   As the community indulges more in the healthy competition of _transformers vs ConvNets_ and
   ponders whether self-attention is even necessary, the question posed in this paper is well-timed.
   The paper motivates readers and practitioners to rethink the reason behind recent successful vision models.

2. **Simple model architecture:**
   ConvMixer follows an extremely simple fully-convolutional design.
   The authors start with a design philosophy of enabling _mixing_ of information across spatial and channel dimensions,
   and achieve it solely by convolution layers (depthwise and pointwise, respectively).
   Depthwise convolution is much simpler to understand and implement than self-attention.
   Moreover, the entire model architecture can be expressed and specified using only four scalar parameters.

3. **Efficient model architecture:**
   ConvMixer has convolutional layers that operate in patches.
   Depthwise convolutions do not incur quadratic computation costs like self-attention since they have "sliding window" computation.
   This also carries a few other subtle benefits: first, this model does not require position embeddings (like MLP-Mixer).
   Second, it also enables easy inference and transfer learning for tasks that require a different resolution
   (e.g. object detectors use much higher resolution) without needing additional tricks such as bilinear interpolation of patch embeddings.

4. **Strong empirical performance:**
   The authors state that the reported results are obtained by using sensible defaults of the ImageNet-1k training recipe.
   It is encouraging to see the strong empirical performance of ConvMixer despite limited hyperparameter search.
   Comparisons with recent model architectures in the main paper are sufficient; more are included in Appendix A.
   I side with the authors' proposition that the goal here is to not show state-of-the-art performance.
   With this in mind, I believe the performance trends matter more than absolute numbers, and they are sufficient.

5. **Neat writing clarity and presentation:**
   I believe that the overall writing quality and presentation in this paper are excellent.
   The authors have done due diligence in extensively discussing relevant literature.
   The model architecture is visually illustrated neatly with a self-contained caption
   -- it aids the reader in quickly understanding the structure of ConvMixer.
   Each table that presents quantitative results, has a single takeaway message.
   The authors provide a clean source code, and an even more concise _tweet-worthy_ implementation in Appendix E.
   I tried it and it successfully instantiates a PyTorch module.

**Additional remark on contributions and impact:**
Overall, I am comfortable recommending this paper for acceptance, without further changes.
I agree with the authors' proposition — the goal here is to not introduce state-of-the-art architecture.
As a member of the research community, I am well aware that an earlier version of this paper has been openly accessible for more than one year.
The paper has gained 100+ citations already — the community has already realized its positive impact.
This paper was also in the limelight for its short length and its bold and unconventional writing style;
I would not gate-keep simply because the writing style does not ~confer~ (EDIT typo:) conform to my preference,
so long as it does not violate any official policy (to the best of my knowledge, it does not).
Given its technical strengths and already observed impact, I think this paper is ready to be accepted at TMLR.

### Weaknesses:

I found a few minor weaknesses that I list here for completeness.
I don't feel any of them to be critical enough for recommending rejection.

1. There is one main weakness that underpins even some of my listed strengths —
   the authors could further strengthen their contributions by presenting ConvMixer as a powerful architecture even at a larger scale.
   This would include more thorough tuning of the model architecture, optimization, and presenting results with even larger models (compared to the size of ViT-L/H).
   The authors may have limited computing resources, so this weakness is not critical to address.

2. The authors claim that ConvMixer can work at any resolution (as listed in strengths).
   While valid by design, it is worthwhile to show inference speed and accuracy trends when a trained ConvMixer is employed for inference at varying image resolutions.

3. ConvMixer uses BatchNorm by default, instead of LayerNorm as used in ViT.
   The authors include an ablation showing that LayerNorm works well but incurs a very small performance drop.
   I suggest recommending LayerNorm as a default as it allows seamless batch size scaling across GPUs (without complicated BatchNorm synchronization across devices).
   The authors may consider ConvNeXt paper for gaining some intuition on architectural changes that can facilitate the use of LayerNorm instead of BatchNorm effortlessly.

---

> ### Author Response · Authors · 2023-02-16
> **Response**
>
> Thanks for your review! You summarized the point of our paper very nicely, which we are glad to see.
>
> Strengths
>
> 5. Thank you; our hope was that the compact, concise paper format would actually be easier to read and understand than a longer paper.
>
> Weaknesses
>
> 1. We agree this is a weakness and that it would be great to investigate larger ConvMixers trained on larger datasets (e.g., ImageNet-21k or even JFT-scale datasets). However, we simply don’t have the compute for this investigation. And the largest vision datasets are not even public. To some extent, work published after our paper shows that a model quite similar to ConvMixer scales well on ImageNet-21k [1]. But even then, one can still ask if it scales further (e.g., to JFT-300M).
>
> 2. That’s a good suggestion, thanks. We will include this in the paper soon and let you know when we have updated it.
>
> 3. We concede that it may have made more sense to use LayerNorm than BatchNorm in ConvMixer. However, as you stated, the paper has been openly accessible for a long time and has been extended in various ways by various papers. At this point, we think that BatchNorm is essentially “locked in”, even if it may not be the most ideal choice (i.e., it is presumably established that the architecture widely-known as “ConvMixer” uses BatchNorm). It does make sense, of course, to provide results and weights for a large-scale ConvMixer using LayerNorm – we will eventually get around to adding this to the paper (and GitHub repos), but in the near future we can’t spend our limited compute resources on this.
>
> [1] https://arxiv.org/abs/2201.03545

---

> > ### Comment · Reviewer_9PYK · 2023-02-16
> > **Convinced by the response.**
> >
> > Thank you for your response! I already stated that all my stated weaknesses are not critical enough to warrant a rejection. I stand by my earlier assessment — all the above responses are sufficient. Replying to individual comments:
> >
> > 1. Glad we are on the same page here. I understand that this study requires a lot of computational resources, but I am happy that the current version of the paper can stand on its own without this study. The evidence of scaling in [1] is convincing.
> >
> > 2. Thank you, glad you like the suggestion! I am eager to see the revised paper with new results!
> >
> > 3. I appreciate that you are receptive to my suggestion, and I also understand that the prevalence of ConvMixer in follow-up works has "locked in" the default architecture. It is okay to not include LayerNorm results in the paper for the time being — the choice of normalization layer does not alter the central message in the paper anyway.

---

> > > ### Author Response · Authors · 2023-02-19
> > > **Additional plot**
> > >
> > > We're just writing to let you that we added your suggested plot the paper (Figure 4 in the appendix), as well as a short associated paragraph at the end of Appendix B. Please let us know what you think, and thanks again.

---

> > > > ### Comment · Reviewer_9PYK · 2023-02-22
> > > > **Thank you for including the plot.**
> > > >
> > > > I think the new plot in the appendix provides sufficient detail, thank you for including it!

---

### Author Response · Authors · 2023-02-19
**Summary of responses, changes**

Thanks again to all the reviewers for their helpful feedback and comments.

The reviewers suggested several changes to the paper, though we were only able to include one in our revision: the suggestion to include a graph of inference time versus input size (Figure 4 in the appendix), since compatibility with variable input sizes is one of the advantages of ConvMixer. We added an associated paragraph at the end of Appendix B.

The other main suggested changes were not feasible due to our compute budget: scaling up ConvMixers (in terms of the number of parameters and data set size), re-running our main experiments with LayerNorm instead of BatchNorm, and comparing ConvMixer to other models after hyperparameter optimization (which we did not do originally). While we agree these are all good suggestions and hope to eventually complete them in the future, it simply is not feasible for us currently.

Overall, we were pleased to see that the main points of our paper were sufficiently clear despite the slightly unconventional format.

We will update this post in the event of any additional discussion in the next couple of days.

---

### Decision · Action_Editors · 2023-03-07

**Recommendation:** Accept as is

**Comment:**

Meta Review for “Patches Are All You Need?”

As the three reviewers (9PYK, 7eQ2, p1Lh) wrote, the authors of "Patches Are All You Need?" pose an interesting question for the computer vision community: whether the success of recent Vision Transformers and MLP-Mixers can be largely attributed to processing images as patches. Instead of following the trend in deep learning papers that propose ever more complex architectures, this work investigates a single, simple and meaningful goal of understanding the effect of using patches and finding most efficient ways of processing them. To do this, the authors consider the recent success of Vision Transformers and other recent computer vision models such as MLP-Mixer, and pose a highly simplified architecture.

The resulting architecture, ConvMixer, is simple compared to prior work, both conceptually and in terms of the number of parameters, yet the authors show it achieves fairly strong performance on ImageNet even without many optimizations. To synthesize the expression of simplicity, the authors have made great efforts to summarize the message of their work in 6 pages, which hopes to send a message to the community to emphasize the value of being concise in scientific communication.

Given the technical strengths and presentation quality, all reviewers, and myself recommend acceptance. This work is relevant to all deep learning practitioners and researchers working with CNNs, and we expect this work to have a good impact even beyond the computer vision sub-field. Lastly, the paper sets the standard for the community to follow for presenting ideas elegantly in a succinct manner.

**Audience:**

See meta-review below in "comments".

**Claims And Evidence:**

See meta-review below in "comments".